# Dissection of goadsporin biosynthesis by *in vitro* reconstitution leading to designer analogues expressed *in vivo*

Taro Ozaki[1,*,†], Kona Yamashita[1,*], Yuki Goto[2], Morito Shimomura[1], Shohei Hayashi[1,†], Shumpei Asamizu[1], Yoshinori Sugai[1], Haruo Ikeda[3], Hiroaki Suga[2] & Hiroyasu Onaka[1,4]

Goadsporin (GS) is a member of ribosomally synthesized and post-translationally modified peptides (RiPPs), containing an N-terminal acetyl moiety, six azoles and two dehydroalanines in the peptidic main chain. Although the enzymes involved in GS biosynthesis have been defined, the principle of how the respective enzymes control the specific modifications remains elusive. Here we report a one-pot synthesis of GS using the enzymes reconstituted in the 'flexible' *in vitro* translation system, referred to as the FIT–GS system. This system allows us to readily prepare not only the precursor peptide from its synthetic DNA template but also 52 mutants, enabling us to dissect the modification determinants of GodA for each enzyme. The *in vitro* knowledge has also led us to successfully produce designer GS analogues *in vivo*. The methodology demonstrated in this work is also applicable to other RiPP biosynthesis, allowing us to rapidly investigate the principle of modification events with great ease.

[1] Department of Biotechnology, Graduate School of Agricultural and Life Sciences, The University of Tokyo, Bunkyo-ku, Tokyo 113-8657, Japan. [2] Department of Chemistry, Graduate School of Science, The University of Tokyo, Bunkyo-ku, Tokyo 113-0033, Japan. [3] Kitasato Institute for Life Sciences, Kitasato University, Sagamihara, Kanagawa 252-0373, Japan. [4] Biotechnology Research Center, Toyama Prefectural University, Imizu, Toyama 939-0398, Japan. * These authors contributed equally to this work. † Present addresses: Department of Chemistry, Graduate School of Science, Hokkaido University, Sapporo, Hokkaido 060-0810, Japan (T.O.); Department of Agricultural and Forest Sciences, Faculty of Life and Environmental Science, Shimane University, 1060 Nishikawatsu, Matsue, Shimane 690-8504, Japan (S.H.). Correspondence and requests for materials should be addressed to H.S. (email: hsuga@chem.s.u-tokyo.ac.jp) or to H.O. (email: aonaka@mail.ecc.u-tokyo.ac.jp).

Goadsporin (GS) is a peptidic natural product having an N-terminal acetyl group, six azoles and two dehydroalanines (Dha), produced by *Streptomyces* sp. TP-A0584 (ref. 1). It exhibits potent antibacterial activity against actinomycetes, including *Streptomyces scabies* JCM7914, a plant pathogen that causes the disease potato scab[2]. Although the three-dimensional structure is unknown, GS very likely crosses the cell membrane and targets signal recognition particle that plays a critical role in proper cellular localization of nascent proteins[3]. Moreover, it has been shown that GS is capable of eliciting secondary metabolism and promoting morphogenesis in various actinomycetes[2]. Therefore, GS is an attractive scaffold for the development of not only novel antibiotics with unique modes of action but also unique chemical tools inducing hidden metabolic capacities in actinomycetes.

Biosynthetically, GS belongs to a family of ribosomally synthesized and post-translationally modified peptides (RiPPs)[4], which are generally produced via ribosomal synthesis of a precursor peptide and subsequent post-translational modifications (PTMs). The GS biosynthetic gene cluster contains a structural gene *godA* and six PTM enzyme genes (*godB, D–H*)[3]. The precursor peptide GodA is composed of an N-terminal 30 amino-acid leader peptide (LP) required for recognition by the PTM enzymes and a C-terminal 19 amino-acid core peptide that matures into GS. Some of the Ser, Thr and Cys (S/T/C) residues present in the core peptide region are firstly modified to the corresponding azolines by the cyclodehydratase[5–9] GodD and subsequently oxidized to azoles by the flavin mononucleotide-dependent dehydrogenase[5,10,11] GodE. Moreover, some Ser residues are converted to Dha residues via O-glutamylated serine cooperatively by GodF and GodG that are homologous to the N-terminal and C-terminal domains of the LanB dehydratase, respectively[12–14]. The N-terminal LP is likely removed by GodB, which is a putative ABC transporter bearing an N-terminal peptidase domain. After removal of the LP, the biosynthesis is completed with acetylation of the N-terminal amino group catalysed by the GNAT domain-containing acetyltransferase GodH[15,16].

Since the PTM enzymes involved in RiPPs biosynthesis pathways often tolerate minor alterations of the substrate sequences, some analogues of RiPPs have been successfully expressed by engineering of the wild-type precursor peptide genes in appropriate host cells[17–20]. However, investigation on the substrate tolerance of each PTM enzyme or cooperative PTM enzymes often requires the laborious preparation of mutated substrate genes and optimization of their expression in the host cells; even if these steps are achieved, PTM enzymes could fail to modify the mutant substrates for undefined reasons, which makes it difficult to draw conclusions from such studies. With regard to the biosynthesis of GS, we have not yet witnessed comprehensive investigations of the substrate tolerance of the PTM enzymes.

We previously devised an *in vitro* reconstituted biosynthetic system[21] combination of a custom-made cell-free translation (flexible *in vitro* translation; FIT[22]) system composed of reconstituted translation components[23] with the post-translational cyclodehydratase PatD[9]. This system, referred to as the FIT–PatD system, enabled the expression of a variety of precursor peptides from synthetic DNA templates and subsequent PatD-catalysed cyclodehydration of S/T/C residues in a one-pot manner, facilitating analysis of the substrate tolerance of PatD. Extensive mutagenesis of the precursor peptides using the FIT–PatD system revealed the critical substrate recognition determinants in the leader, recognition and core peptide sequences of the substrate, showing unprecedented tolerance of the PatD reaction *in vitro*. Although this particular work exploited only the *in vitro* activity of PatD, it shows the advantage of rapid investigation of this particular PTM enzyme.

Here we report a one-pot synthesis of GS using reconstituted biosynthesis machinery, coupled with the FIT system, referred to as the FIT–GS system. This system enables us to produce native GS from the corresponding synthetic DNA template. Moreover, it allows for omitting desired PTM enzymes in GS biosynthesis (GodPTMs) and thus rapidly investigating the biosynthetic pathway. Most importantly, by means of the FIT–GS system, various GodA mutants could be readily expressed. On the basis of such *in vitro* knowledge, we were able to produce designer GS derivatives *in vivo* as well as *in vitro*, whose structures were determined by mass spectrometry (MS) fragmentation and/or nuclear magnetic resonance (NMR).

## Results

**Preparation of the precursor peptide, GodA\***. To establish the FIT-GS system (Fig. 1a,b), a synthetic DNA template encoding an engineered GodA (GodA\*, Supplementary Fig. 1) was designed and synthesized by primer extension followed by PCR. To improve the ionization efficiency in mass spectrometry, three successive lysines were introduced between Met-30 and Glu-29 of the native GodA[24]. GodB, a putative peptidase cleaving out the LP region of GodA, contains a transmembrane domain and could not be expressed as a soluble protein. Therefore, a glutamic acid residue was added at the −1 position of the core peptide to enable the removal of the LP by a commercial protease, GluC. Following the *in vitro* ribosomal synthesis of GodA\* in the FIT system, a signal representing *m/z* 5,482 was identified by matrix-assisted laser desorption/ionization–time of flight–MS (MALDI–TOF–MS) analysis (Fig. 2a); this signal corresponds to GodA\* with an intramolecular disulfide bond between two Cys residues in the core peptide region. GodA\* was used in the presence of 1 mM dithiothreitol (DTT) to reduce the disulfide bond in the following studies with the reconstituted GodPTM enzymes.

**Preparation of the recombinant GodPTM enzymes**. The GS biosynthetic enzymes were expressed as recombinant proteins in *Escherichia coli*. Recombinant GodD, GodE, GodF and GodH were successfully expressed as soluble proteins (Supplementary Fig. 2). Unfortunately, GodG, a putative glutamate elimination enzyme, could not be obtained as an active enzyme. Instead of GodG, we conceived that the function of GodG could be replaced by that of a homologous enzyme, LazF, originating from lactazole biosynthesis[25]. LazF is a chimeric protein composed of an N-terminal glutamate elimination domain and a C-terminal dehydrogenase domain (Supplementary Fig. 3), and was successfully expressed as a soluble protein in *E. coli* (Supplementary Fig. 2). We observed that recombinant GodE and LazF exhibited a yellow colour, which is an indication that binding to a flavin cofactor had occurred. Indeed, the liquid chromatography (LC)–MS analysis of methanol extracts of these two proteins revealed that flavin mononucleotides were non-covalently bound to these enzymes (Supplementary Fig. 4). On the other hand, recombinant GodH was co-purified with acetyl-CoA (Supplementary Fig. 5), consistent with its putative function (*N*-acetyltransferase).

**Establishing the FIT–GS system**. To establish the FIT–GS system, we first included GodD in the FIT system to see if cyclodehydration of the expressed GodA\* could occur. We observed four major signals at *m/z* 5,448, 5,430, 5,412 and 5,394 (Fig. 2b), which were assigned to the GodA\* peptides

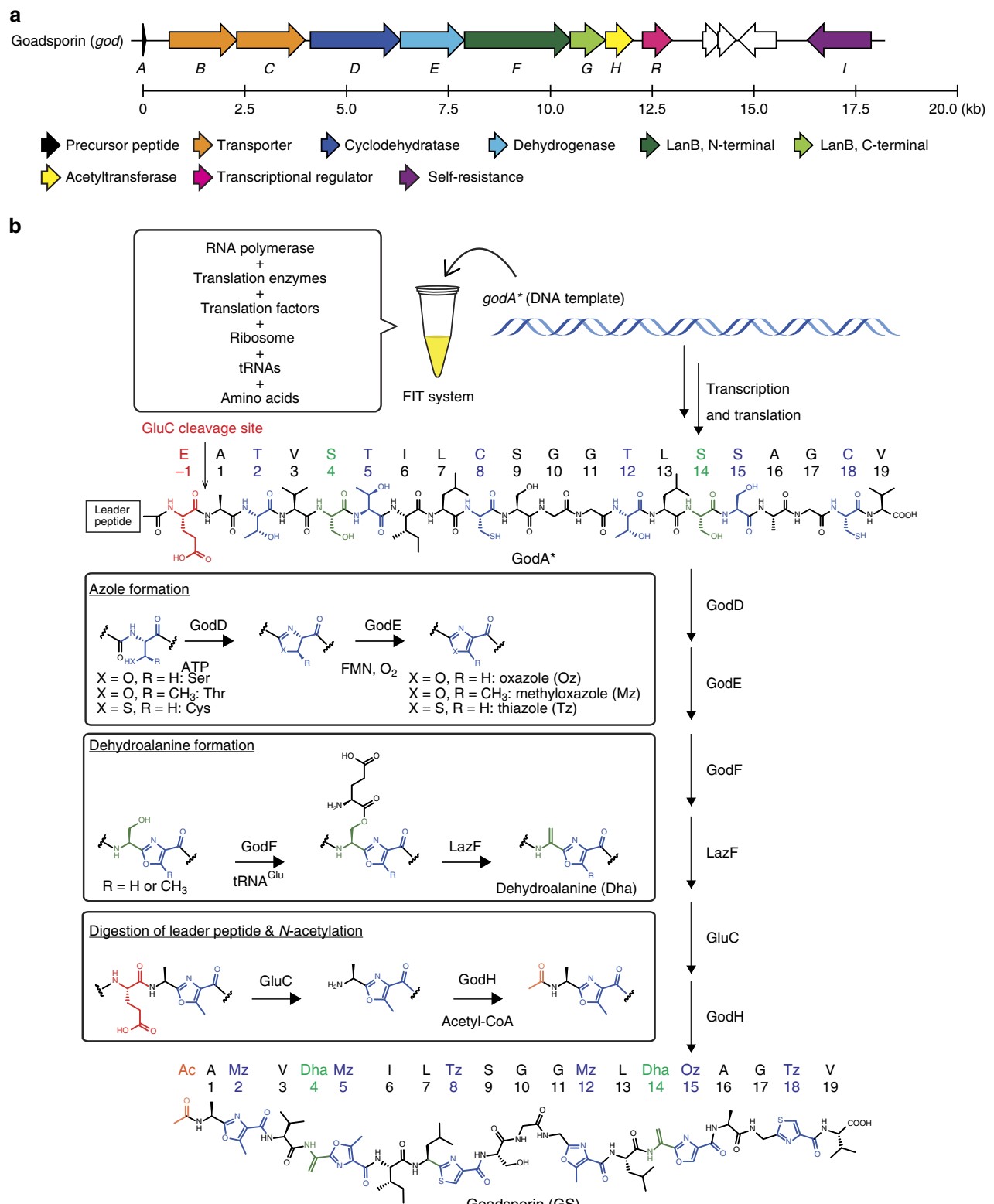

**Figure 1 | The overview of an enzymatic total synthesis of GS by the FIT–GS system. (a)** The biosynthetic gene cluster for GS. **(b)** GodA*, an engineered precursor peptide, is synthesized by the FIT system and converted into GS by six PTM enzymes.

decorated with 2, 3, 4 and 5 azoline moieties, respectively. Surprisingly, when GodE was also included (the FIT–GodD/E system), the above cyclodehydrated signals were converted to a single signal at $m/z$ 5,364, which was unmistakably assigned to GodA* containing six azoles (Fig. 2c). The data indicated that

the GodA* could undergo incorporation of six azoles in the presence of GodD and GodE.

We next attempted to install two Dha residues into GodA*. We have previously proposed that the two Dha residues could be biosynthesized via GodF-catalysed glutamylation using

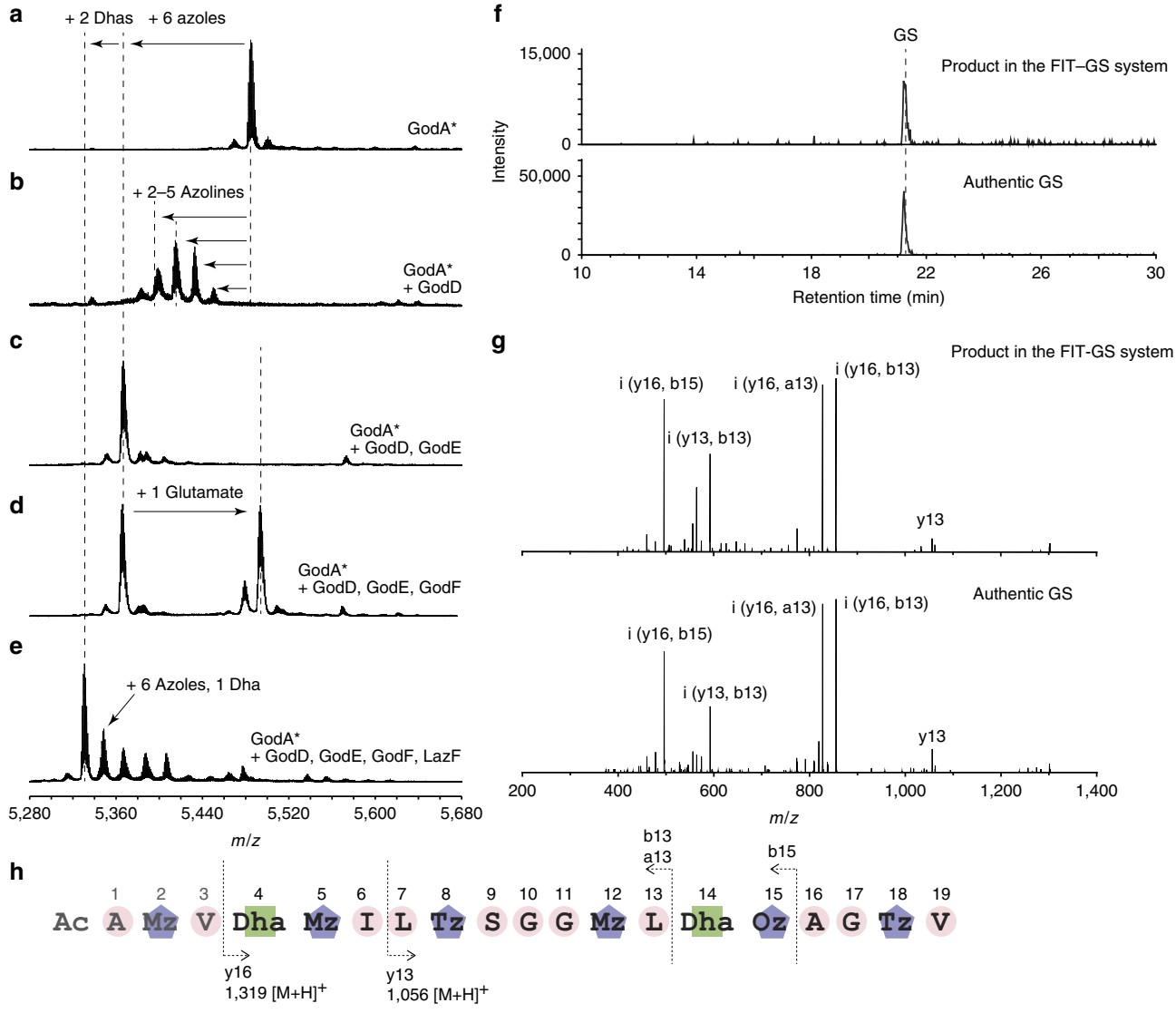

**Figure 2 | MS analysis of the enzymatic reactions.** (**a–e**) MALDI–TOF–MS spectra of GodA* expressed in the FIT system (**a**), reaction products of FIT–GodD system (**b**), FIT–GodD/E system (**c**), FIT–GodD/E/F system (**d**) and FIT–GodD/E/F/LazF system (**e**) are shown. (**f**) Total ion current (TIC) of LC–MS$^3$ analysis of the reaction product of the FIT–GS system and the authentic GS (62.5 pg of authentic GS was injected for this chromatogram). (**g**) MS$^3$ spectrum of the authentic GS and GS synthesized in the FIT–GS system. (**h**) Observed MS$^3$ fragmentation pattern of GS. Ac, acetyl group; Dha, dehydroalanine; Mz, methyloxazole; Oz, oxazole; Tz, thiazole. Calculated and observed $m/z$ values in MALDI–TOF–MS analyses are summarized in Supplementary Data.

Glu-transfer RNA (tRNA)$^{Glu}$ as a co-substrate and subsequent GodG-catalysed elimination of the glutamate based on the result of gene disruption experiments[13]. When GodF was incubated with the FIT-expressed GodA* in the presence of GodD/E (the FIT–GodD/E/F system), a signal at $m/z$ 5,493 was observed, which was consistent with the GodA* comprising six azoles and one glutamyl group (Fig. 2d). Because *E. coli* tRNA$^{Glu}$ exists in the FIT system, this co-substrate was probably used for the GodF-catalysed glutamylation. The glutamylated products were not observed when GodF was solely incubated with GodA* (Supplementary Fig. 6), suggesting that the azoles installed in GodA* by GodD/E are required for substrate recognition by GodF.

As described earlier, since GodG could not be expressed in an active form, we next attempted to supplement the system with LazF. When LazF was included in the FIT–GodD/E/F system, GodA* was converted to a product whose peak was observed at $m/z$ 5,328 (Fig. 2e). This $m/z$ value is consistent with the expected value of GodA* bearing six azoles and two Dha residues. This indicates that even though the LP sequence in GodA* is not the same as that in lactazole precursor peptide, LazF was able to eliminate the glutamyl group from the *O*-glutamylated serine residues in GodA* to form the Dha groups.

To complete the reconstitution of the *in vitro* GS biosynthesis machinery, GluC and GodH were included in the FIT–GodD/E/F/LazF system. The endoprotease GluC should be capable of cleaving the peptide bond between the glutamic acid residue (Glu-1) and the following core peptide, and then the liberated N-terminal amino group would be acetylated by GodH. On the inclusion of GluC and GodH, the product was analysed by a selective LC–MS/MS/MS (MS$^3$) method, which was established for the analysis of naturally occurring GS. We observed an LC profile and the fragment pattern of the MS$^3$, which matched completely with those observed for the authentic GS (Fig. 2f–h).

**Figure 3** shows a table titled "Amino-acid sequences of core peptides" with positions 1–19.

| Entry | Name | Core peptide sequence (positions 1–19) |
|---|---|---|
| 1 | GodA* | A T V S T I L C S G G T L S S A G C V |
| 2 | GodA*1–9 | A T V S T I L C S |
| 3 | GodA*1–6 | A T V S T I |
| 4 | GodA*1–3 | A T V |
| 5 | GodA*1–2 | A T |
| 6 | GodA*2–3 | (2) T V |
| 7 | GodA*1–9-T2A/T5A/C8A | A A V S A I L A S |
| 8 | GodA*1–6-T2A/T5A | A A V S A I |
| 9 | GodA*1–3-T2A | A A V |
| 10 | GodA*1–8 | A T V S T I L C |
| 11 | GodA*2–9 | (2) T V S T I L C S |
| 12 | GodA*1–5 | A T V S T |
| 13 | GodA*2–6 | (2) T V S T I |
| 14 | GodA*1–3-A1C/T2A | C A V |
| 15 | GodA*1–3-A1T/T2A | T A V |
| 16 | GodA*1–3-A1S/T2A | S A V |
| 17 | GodA*1–3-T2A/V3C | A A C |
| 18 | GodA*1–3-T2A/V3T | A A T |
| 19 | GodA*1–3-T2A/V3S | A A S |
| 20 | GodA*1–9-A1T/T2A | T A V S T I L C S |
| 21 | GodA*1–6-A1T/T2A | T A V S T I |
| 22 | GodA*1–6-T5I/I6T | A T V S I T |
| 23 | GodA*1–9-T2V/V3T | A V T S T I L C S |
| 24 | GodA*1–6-T2V/V3T | A V T S T I |
| 25 | GodA*1–9-L7C/C8L | A T V S T I C L S |
| 26 | GodA*1–9-$_{-1}$A$_1$ | A A T V S T I L C S |
| 27 | GodA*1–9-$_3$A$_4$ | A T V A S T I L C S |
| 28 | GodA*1–9-$_6$A$_7$ | A T V S T I A L C S |
| 29 | GodA*1–3-T2C | A C V |
| 30 | GodA*1–3-T2S | A S V |
| 31 | GodA*1–3-A1E | E T V |
| 32 | GodA*1–3-A1R | R T V |
| 33 | GodA*1–3-A1Y | Y T V |
| 34 | GodA*1–3-A1W | W T V |
| 35 | GodA*1–3-V3R | A T R |
| 36 | GodA*1–3-V3Y | A T Y |
| 37 | GodA*1–3-V3W | A T W |
| 38 | GodA*1–3-V3E | A T E |

Legend: ● Unmodified residue; ⬠ (purple pentagon) Azole; ⬠ (open pentagon frame) Azoline.

**Figure 3 | Summary of azole formation in the FIT–GodD/E system.** The core peptide regions of the constructed GodA* derivatives expressed in the FIT–GodD/E system for mutation studies in azole formation are shown. Residues modified to azoline and azole are highlighted with purple pentagonal frames and purple pentagons, respectively. Unmodified residues in core peptide regions are highlighted with cherry pink circles. Mutated residues are coloured red.

The observed peak areas of GS expressed in the FIT–GS system allowed for estimating a concentration to be an ∼7 nM in comparison with known quantities of the authentic GS samples. These results explicitly showed the *in vitro* synthesis of the desired GS in the FIT–GS system.

**Mutation studies of GodA to probe azole formation.** As it is possible to quickly generate any mutant *godA** DNA sequence, we envisaged that the FIT–GS system could readily examine the substrate tolerance of GodD/E for the installation of azoles. We first designed five DNA templates that expressed a series of GodA* deletion mutants, GodA*1–9, GodA*1–6, GodA*1–3, GodA*1–2 and GodA*2–3 (Fig. 3, entries 2–6, the number corresponds to the residue in GodA* shown in Fig. 3, entry 1). Using the FIT–GodD/E system, GodA*1–9, GodA*1–6 and GodA*1–3 were converted to the expected tri-, di- and mono-azole products, respectively (Fig. 4a–c). On the other hand, we observed no modification on GodA*1–2 and GodA*2–3,

indicating that these mutants have lost important recognition elements for the GodD/E-catalysed modification (Supplementary Fig. 7). Since GodA*1–9 and GodA*1–6 contain potential modification sites at S4 and S9 in addition to canonical T2, T5 and C8, we prepared the alanine mutants of GodA*1–9-T2A/T5A/C8A and GodA*1–6-T2A/T5A (Fig. 3, entries 7 and 8). We observed peaks for the respective intact peptides expressed in the FIT–GodD/E system (Supplementary Fig. 8). Likewise, the GodA*1–3-T2A mutant was not modified (Supplementary Fig. 8; Fig. 3, entry 9). We thus concluded that the azoles were correctly installed at the expected T/C residues in these deletion mutants.

We wondered if the adjacent residue(s) to the modifying S/T/C site is important for azole formation. To investigate this, two deletion mutants of GodA*1–9, GodA*1–8 and GodA*2–9, were expressed in the FIT–GodD/E system (Supplementary Fig. 9a,b; Fig. 3, entries 10 and 11). Although each mutant had three potential modification sites, we observed products with only two azoles. Likewise, the two deletion mutants of GodA*1–6,

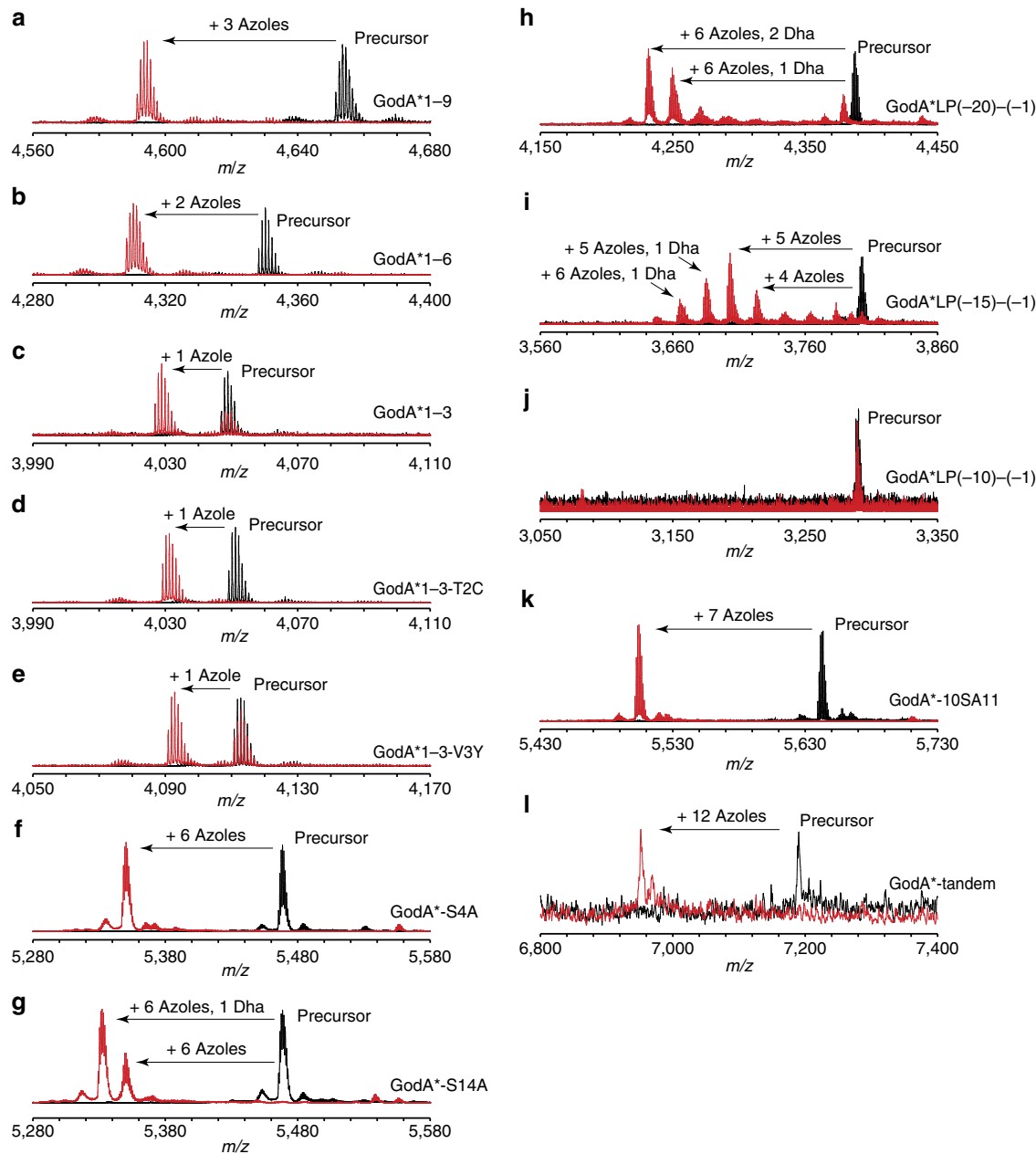

**Figure 4 | Mutagenesis study of GodA\* by means of the FIT–GS system.** (a–e) Azole formation in the FIT–GodD/E system was investigated with GodA\*1–9 (**a**), GodA\*1–6 (**b**), GodA\*1–3 (**c**), GodA\*1–3-T2C (**d**) and GodA\*1–3-V3Y (**e**). (**f,g**) The order of Dha formation was investigated with GodA\*-S4A (**f**) and GodA\*-S14A (**g**) using the FIT–GodD/E/F/LazF system. (**h–j**) Sequence requirement of the LP region in the FIT–GodD/E/F/LazF system was investigated with GodA\*LP(−20)-(−1) (**h**), GodA\*LP(−15)-(−1) (**i**) and GodA\*LP(−10)-(−1) (**j**). (**k,l**) noncanonical azole formation on the designer GodA\* analogue in the FIT–GodD/E system was investigated with GodA\*-10SA11 (**k**) and GodA\*-tandem (**l**). Calculated and observed m/z values in MALDI-TOF-MS analyses are summarized in Supplementary Data.

GodA\*1–5 and GodA\*2–6, were converted to the mono-azole products, instead of the expected two azoles (Supplementary Fig. 9c,d; Fig. 3, entries 12 and 13). As described earlier, neither Thr residue in GodA\*1–2 nor GodA\*2–3 was modified in the FIT–GodD/E system (Fig. 3, entries 5 and 6). Taken together, the results allowed us to hypothesize that a motif of $X_1$-(S/T/C)-$X_2$ could be required for the formation of azoles.

To confirm this hypothesis, we first introduced double mutations into GodA\*1–3, where the position of the S/T/C residue was altered from the original motif, that is, (S/T/C)-$X_1$-$X_2$ or $X_1$-$X_2$-(S/T/C; Fig. 3, entries 14–19). As expected, neither mutant was modified in the FIT–GodD/E system (Supplementary

Fig. 10a–f). We also prepared similar double mutants of GodA\*1–9 and GodA\*1–6, as well as GodA\*1–3 (Fig. 3, entries 20–25). The swapped S/T/C residues in these mutants were not modified at all, except for GodA\*1–9-L7C/C8L, where the L7C residue was converted to azoline, not azole (Supplementary Fig. 11a–f).

Although the above result supports our hypothesis that the $X_1$-(S/T/C)-$X_2$ motif is the determinant of azole formation by GodD/E, it is possible that GodD/E would strictly recognize the positions of S/T/C residues in the core peptide region of GodA. To rule out this possibility, we prepared three insertion mutants of GodA\*1–9, in which a single alanine was embedded

| Entry | Name | Amino-acid sequences of core peptides |
|-------|------|---------------------------------------|
| 1 | GodA* | A T V S T I L C S G G T L S S A G C V |
| 2 | GodA*1–16 | A T V S T I L C S G G T L S S A |
| 3 | GodA*1–13 | A T V S T I L C S G G T L |
| 4 | GodA*1–9 | A T V S T I L C S |
| 5 | GodA*1–6 | A T V S T I |
| 6 | GodA*-T5A | A T V S A I L C S G G T L S S A G C V |
| 7 | GodA*-S15A | A T V S T I L C S G G T L S A A G C V |
| 8 | GodA*-S4A | A T V A T I L C S G G T L S S A G C V |
| 9 | GodA*-S14A | A T V S T I L C S G G T L A S A G C V |

○ Unmodified residue ⬠ Azole ▪ Dehydroalanine (Dha)

**Figure 5 | Summary of Dha formation in the FIT–GodD/E/F/LazF system.** The core peptide regions of the constructed GodA* derivatives expressed in the FIT–GodD/E/F/LazF system for mutation studies in Dha formation are shown. Residues modified to azole and Dha are highlighted with purple pentagons and green squares, respectively. Unmodified residues in core peptide regions are highlighted with cherry pink circles. Mutated residues are coloured red.

at the first, fourth or seventh position (Fig. 3, entries 26–28). Being consistent with our hypothesis, these mutants were fully modified to give the expected tri-azole products (Supplementary Fig. 12). Thus, we conclude that the $X_1$-S/T/C-$X_2$ motif is the recognition determinant for the azole formation catalysed by GodD/E.

We next addressed a question whether GodD/E could tolerate variations of the $X_1$-(S/T/C)-$X_2$ motif in GodA*. We used the minimal substrate, GodA*1–3, as a scaffold and introduced point mutations. We first tested the substitution of T2 to C or S (Fig. 3, entries 29 and 30), revealing that these were fully converted to the corresponding azoles (Fig. 4d; Supplementary Fig. 13). We then prepared four mutants, where the A1 residue was substituted with E, R, Y or W (Supplementary Fig. 14a–d; Fig. 3, entries 31–34), and found that the T2 residue in these mutants was unable to be converted to azole. We also prepared four mutants where the V3 residue was substituted with E, R, Y or W. The T2 residue in three out of the four mutants was converted to azole (Fig. 4e; Supplementary Fig. 14e,f; Fig. 3, entries 35–37), whereas that in the V3E mutant (GodA*1–3-V3E) was unmodified (Supplementary Fig. 14g; Fig. 3, entry 38). Taken together, GodD/E do not accept the altered $X_1$ residue in the $X_1$-(S/T/C)-$X_2$, but fairly well tolerates various $X_2$ residues except for an acidic residue.

**Mutation studies on GodA in Dha formation.** The mature GS molecule includes two Dha residues. Since the Dha formation was often found in other RiPPs syntheses, it would be intriguing to reveal how the PTM events in GS biosynthesis are controlled, and how this compares with other systems. We utilized the FIT–GodD/E/F/LazF system and first tested a series of truncation mutants to observe the Dha formation (Fig. 5, entries 2–5). Surprisingly, GodF/LazF failed to modify all of the tested truncated mutants, GodA*1–16, GodA*1–13, GodA*1–9 and GodA*1–6, suggesting that the entire core peptide could be required for the Dha formation (Supplementary Fig. 15b–e). This has led us to further test two Ala point mutants of GodA*, T5A or S15A (Fig. 5, entries 6 and 7). Since the azole installation at these positions was dismissed, these mutants would tell us the importance of the azole residue at these positions for the Dha formation in GodA*. Again, these mutants were not the substrate for GodF/LazF (Supplementary Fig. 16), concluding that the azoles at the 5th and 15th positions

could be essential for the installations of the Dha residues at S4 and S14.

On the basis of the above result, we designed two mutants in which S4 or S14 were mutated to an alanine. When the upstream Dha formation by GodF/LazF was missing in the S4A mutant, we were unable to detect the downstream Dha installation at S14 (Fig. 5, entry 8; Fig. 4f). In contrast, the Dha formation occurs at S4 in the S14A mutant (Fig. 5, entry 9; Fig. 4g). Thus, the upstream Dha installation at S4 dictates the downstream Dha installation at S14. Collectively, we determined the requirements for the Dha installations at S4 and S14 by GodF/LazF as follows; (1) the azole installations at T5 and S15 are necessary and (2) the upstream Dha installation at S4 is required for the downstream Dha installation at S14, that is, the order of the Dha installations is S4 followed by S14.

**Truncation studies of the LP region in GodA.** The N-terminal LP region of precursor peptides in RiPP biosynthesis generally recruits PTM enzymes to install modifications effectively and selectively[21,26–30]. Therefore, it is likely that the GodA* LP could also play a similar role in recruiting GodPTM enzymes, but it is yet unknown if the entire LP sequence is necessary.

To determine the minimal required region in the GodA LP, a series of five truncated GodA* variants were designed and tested in the FIT–GodD/E/F/LazF system (Fig. 6, entries 2–6). Using the FIT–GodD/E/F/LazF system, GodA*LP(−25)–(−1) and GodA*LP(−20)–(−1) were effectively converted to the fully modified product (Fig. 4h; Supplementary Fig. 17). On the other hand, we observed incomplete modification of GodA*LP(−15)–(−1), giving a peptide decorated with five azoles as a major product, along with some minor byproducts (Fig. 4i). Further truncation, such as GodA* LP(−10)–(−1) and GodA* LP(−6)–(−1), resulted in a complete loss of modification, (Fig. 4j; Supplementary Fig. 17). These results indicated that the C-terminal twenty residues of the GodA* LP region were important for GodPTMs. Since shorter peptides are easier to handle; this finding would facilitate the further production of various precursor analogues in the FIT–GS system.

***In vitro* biosynthesis of rationally designed GS derivatives.** On the basis of the above, we designed an artificial precursor peptide, GodA*-10SA11, which has an insertion of two residues, Ser and Ala, between G10 and G11 (Fig. 7, entry 2). This insertion created

| Entry | Name | Amino-acid sequences of LPs | Number of azoles | Number of Dhas |
|---|---|---|---|---|
| | | −34 −33 −32 −31 −30 −29 −28 −27 −26 −25 −24 −23 −22 −21 −20 −19 −18 −17 −16 −15 −14 −13 −12 −11 −10 −9 −8 −7 −6 −5 −4 −3 −2 −1 | | |
| 1 | GodA* | M K K K E N V Q T L A I D D I E N I D A E V T I E E L S S T N G A E | 6 | 1, 2 |
| 2 | GodA*LP(−25)–(−1) | M K K K           L A I D D I E N I D A E V T I E E L S S T N G A E | 6 | 1, 2 |
| 3 | GodA*LP(−20)–(−1) | M K K K                 I E N I D A E V T I E E L S S T N G A E | 6 | 1, 2 |
| 4 | GodA*LP(−15)–(−1) | M K K K                       A E V T I E E L S S T N G A E | 4, 5, 6 | 1 |
| 5 | GodA*LP(−10)–(−1) | M K K K                             E E L S S T N G A E | 0 | 0 |
| 6 | GodA*LP(−6)–(−1) | M K K K                                   S T N G A E | 0 | 0 |

**Figure 6 | Summary of the LP truncation in the FIT–GodD/E/F/LazF system.** The leader peptide regions of the truncated GodA* expressed in the FIT–GodD/E/F/LazF system are shown. Residues are shown by grey circles.

| Entry | Name | Amino-acid sequences of core peptides |
|---|---|---|
| | | 1 2 3 4 5 6 7 8 9 10 11 12 13 14 15 16 17 18 19 |
| 1 | GodA* | A T V S T I L C S G G T L S S A G C V |
| 2 | GodA*-10SA11 | A T V S T I L C S G A G T L S S A G C V |
| 3 | GodA*-tandem | A T V S T I L C S G G T L S S A G C V |
| | | A T V S T I L C S G G T L S S A G C V |

○ Unmodified residue      ⬠ Azole

**Figure 7 | The designer GodA* derivatives expressed in the FIT–GodD/E system.** The core peptide regions of the designer GodA* derivatives expressed in the FIT–GodD/E system are shown.

a new GSA (as an $X_1$-S/T/C-$X_2$) motif into the wild-type GodA*, potentially leading to the formation of an additional azole. This GodA* analogue was expressed in the FIT–GodD/E system and subjected to MALDI–TOF–MS, confirming the production of a peptide where seven azoles were successfully installed (Fig. 4k).

Since GodA*-10SA11 was modified and relevance of an $X_1$-S/T/C-$X_2$ motif was validated, we conceived that a longer GS analogue could be synthesized in a similar manner. To demonstrate this idea, we designed GodA*-tandem that has a 38-mer core peptide composed of tandemly repeated the original core sequence of GodA and expressed it in the FIT–GodD/E system (Fig. 7, entry 3). As expected, installation of 12 azoles into the GodA*-tandem product was observed (Fig. 4l). Thus, we here have demonstrated that the knowledge obtained from a series of mutagenesis studies were used to express designer GS analogues in the FIT–GodD/E system.

***In vivo* production of designer GS analogues.** We have previously reported a system where the *godA* disruptant of *Streptomyces* sp. TP-A0584 (*ΔgodA*) is complemented by *godA* or its mutants[3] along with overexpression of *godR*[31], a gene encoding the transcriptional activator of the GS biosynthetic gene cluster. Even though the FIT–GS system has given us the advantage of rapidly identifying the critical determinants of GodA* modifications, such knowledge was obtained under the conditions where stoichiometric or excess amounts of PTM enzymes were used, and the conclusions could be limited to *in vitro*. To verify if the *in vitro* knowledge of GodPTMs were applicable to *in vivo* environment, we utilized the above *in vivo* system to express some designer GodA derivatives.

We first validated some 'negative' constructs based on our *in vitro* data of GodA* derivatives. We designed GodA mutants with point substitution of A1S, V3S, I6S, L7S, G10S, G11S, L13S, A16S, G17S and V19S, all of which are predicted to forbid azole formation by GodD/E (Fig. 8, entries 2–12). In fact,

MS fragmentation analysis of the major product in all mutants revealed that the newly introduced Ser residue was intact, while other native S/T/C residues were fully modified (Supplementary Figs 18–28). One exception is that in GodA-L7S we observed a minor product, in which the mutated S7 residue was converted to Dha (Supplementary Fig. 29). Taken together, none of the Ser residues introduced in these mutants underwent the azole formation by the enzymes validating our *in vitro* knowledge that the $X_1$-(S/T/C)-$X_2$ motif is necessary for azole formation.

Next, we tested the *in vivo* expression of two 'positive' construct validated by *in vitro* experiments. GS mutants containing a single substitution (I6Y or V19Y) were successfully produced *in vivo* (Supplementary Figs 30 and 31; Fig. 8, entries 13 and 14). This result has confirmed that the $X_2$ residue of the $X_1$-(S/T/C)-$X_2$ motif can be substituted with some noncanonical residues *in vivo* as predicted by the *in vitro* data.

We then expressed the designer GodA-10SA11 *in vivo* (Fig. 8, entry 15; Fig. 9a). In fact, the fully modified GS-10SA11 was successfully produced in the *ΔgodA* strain (Fig. 9b). From 2.5 mg of the pure compound, we elucidated its structure containing seven azoles by means of NMR (Supplementary Figs 32–38; Supplementary Table 1) and MS/MS ($MS^2$) fragmentation (Fig. 9c,d). In addition, *in vivo* expression of the designer GodA-tandem was also conducted, yielding 2.1 mg of GS-tandem (Fig. 8, entry 16; Fig. 9e); its LC–$MS^2$ analysis confirmed the generation of 12 azoles and 4 Dha residues at the expected sites (Fig. 9f–h). Molecular formulae of GS-10SA11 and GS-tandem were elucidated to be $C_{78}H_{103}N_{21}O_{22}S_2$ and $C_{142}H_{190}N_{38}O_{38}S_4$, respectively, by high-resolution electrospray ionization (HR-ESI)–MS (Supplementary Table 2).

## Discussion

We here developed an *in vitro* reconstituted biosynthesis machinery, the FIT–GS system, which enables one-pot enzymatic synthesis of GS from a synthetic DNA template (Fig. 1b). This *in vitro* system is reminiscent of the nature's strategy for

**Figure 8 | The designer GS analogues expressed *in vivo*.** Residues modified to azole and Dha are highlighted with purple pentagons and green squares, respectively. Unmodified residues are highlighted with cherry pink circles. Mutated residues are coloured red.

biosynthesis of RiPPs expressed from the cognate DNA templates encoding the precursor peptides, including an RNA polymerase, the translation machinery and six PTM enzymes that conduct azole and Dha formation, digestion of the LP region and N-terminal acetylation. Due to the economical reason, this *in vitro* system is suitable for qualitative analysis in small scales, not for quantitative analyses. However, it enables us to access the expression of a wide array of mutants and designer analogues simply by *in vitro* preparation of the corresponding DNA templates. This advantage significantly reduces the labours of chemical synthesis or bacterial expression/isolation of precursor peptides previously described[30,32].

Indeed, in the present work, we tested over 50 analogues in the FIT–GS system and revealed important insights into the azole and Dha formation and the LP recognition in GS biosynthesis (Fig. 10). Since the *in vitro* system utilizes comparatively high concentrations of PTM enzymes to complete all the PTMs of GodA*, it was a concern that the principles found *in vitro* are not necessarily applicable to *in vivo*. We therefore verified the principles extracted from the *in vitro* data by producing 15 GS analogues *in vivo*, including designer GS analogues, GS-10SA11 and GS-tandem. Our consistent results between the *in vitro* and *in vivo* experiments have validated our methodology where an *in vitro* system is used to gain insights into the principle of the PTM enzymatic reactions, which facilitated the design of novel analogues that can be produced *in vivo*.

We found that the $X_1$-S/T/C-$X_2$ motif is an essential recognition element of GodD/E (Fig. 10). According to this principle, we have demonstrated the expression of various lengths of azole-containing peptides consisting of the $X_1$-S/T/C-$X_2$ motif ranging from a single to 12 motifs in the FIT–GS system (Fig. 3, entries 2–4, 10–13, 20–30 and 35–37; Fig. 7, entries 2 and 3). To the best of our knowledge, such short (three residues) azole-containing RiPPs have not yet been discovered in nature. However, the successful *in vitro* expression of such short RiPPs will allow us to investigate not only a possibility of their *in vivo* expression but also their potentials for bioactivities. In addition, the *in vitro* and *in vivo* systems successfully produced GS analogues, where the $X_2$ residue of an $X_1$-S/T/C-$X_2$ motif

was substituted with noncanonical residues, and we have successfully introduced an additional azole group(s) at a noncanonical position(s) based on the above principle (Fig. 7, GodA*-10SA11 and GodA*-tandem; Fig. 8, GS-10SA11 and GS-tandem). Taken together, the present study has demonstrated the GS biosynthetic machinery exhibits broad substrate tolerance and is applicable to generate artificial analogues.

The principle of the GodD/E-catalysing azole installation into the $X_1$-S/T/C-$X_2$ motif turns out to be unique compared with that of known azole/azoline installing enzymes. For instance, the pattern appeared in the naturally occurring substrates of PatD/G in patellamide biosynthesis suggests that azoline/azole would be installed at every second residue. However, the recent *in vitro* study on PatD revealed that azoline installation could occur at various positions and even at successive positions of S/T/C residues[21]. The successive installation of azoles has been also found in naturally occurring systems, such as BamB/C/D in plantazolicin biosynthesis[30] and TbtE/F/G in biosynthesis of thiomuracin[33]. Thus, to the best of our knowledge, the installation of azoles into the $X_1$-S/T/C-$X_2$ motif catalysed by GodD/E represents a unique example of a motif-recognizing azole-installing enzymes. However, we believe that more of these enzymes could be discovered in nature.

In the FIT–GS system, two components that are used in Dha formation in the naturally occurring system have been replaced with noncanonical components. The first noncanonical component is LazF, originating from lactazole biosynthesis in *Streptomyces lactacystinaeus*[25], which replaced the unavailable GodG. Although the N-terminal domain of LazF and GodG, both of which are putatively responsible for glutamate elimination, have a low homology (23% identity), it is rather surprising to see that LazF can cooperatively function with other GS biosynthetic enzymes. This suggests that LazF does not require the interaction with the LP, but likely it acts as a versatile catalyst for eliminating glutamylated S/T to install the Dha residue. The second noncanonical component is *E. coli* Glu-tRNA[Glu] endogenously present in the FIT system instead of *Streptomyces* Glu-tRNA[Glu] for the glutamylation function of GodF. Although GodF is a unique stand-alone enzyme that catalyses glutamylation of

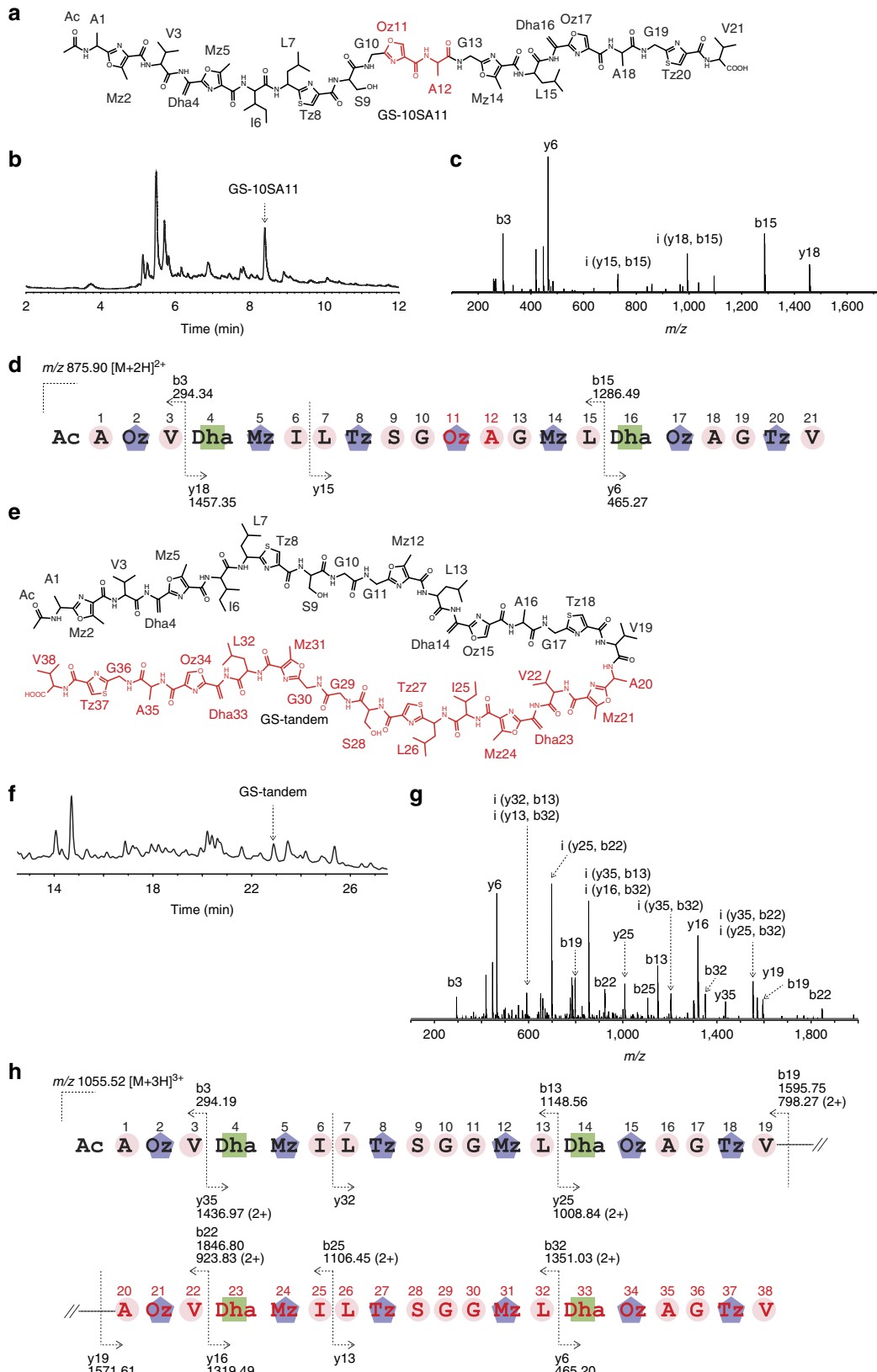

**Figure 9 | *In vivo* production of designer GS analogues.** (**a**) Structure of GS-10SA11. (**b**) HPLC analysis of *n*-butanol extract of *Streptomyces* sp. TP-A0584 *ΔgodA/*pGODR, pTYM-10SA11. The chromatogram was extracted at 254 nm. (**c**) MS$^2$ spectrum of the precursor ion representing *m/z* 876 [M + 2H]$^{2+}$. (**d**) Assignment for the fragmentation pattern of GS-10SA11. (**e**) Structure of GS-tandem. (**f**) HPLC analysis of *n*-butanol extract of *Streptomyces* sp. TP-A0584 *ΔgodA/* pTYM-GS tandem. The chromatogram was extracted at 254 nm. (**g**) MS$^2$ spectrum of the precursor ion representing *m/z* 1,056 [M + 3H]$^{3+}$. (**h**) Assignment for the fragmentation pattern of GS-tandem. Ac, acetyl group; Dha, dehydroalanine; Mz, methyloxazole; Oz, oxazole; Tz, thiazole.

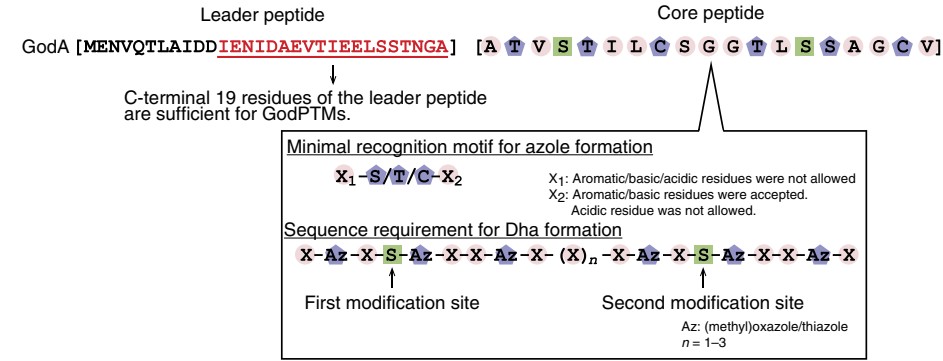

**Figure 10 | Sequence requirements for GodPTMs dissected in this study.** Through examining various analogues both *in vitro* and *in vivo*, sequence requirements for GodPTMs were proposed. The C-terminal 19 residues highlighted in red in the LP region were sufficient for the formation of azoles and Dhas. In the azole formation catalysed by GodD/E, X1-S/T/C-X2 motif could be the recognition determinant. The length of the central region in the core peptide can be varied from one to three. Dha formation required preceding azole formation. Ser4 would be modified before Ser14. Residues modified to azole and Dha are highlighted with purple pentagons and green squares, respectively. Unmodified residues are highlighted with cherry pink circles.

S/T residues, a similar promiscuous usage of the noncognate *E. coli* Glu-tRNA$^{Glu}$ has been seen in a recent study on NisB, which is a dual function enzyme composed of a glutamylation domain and an elimination domain[12]. On the other hand, it has been shown that two other known homologues of NisB, TbtB[33] and MibB[34], could not use *E. coli* Glu-tRNA$^{Glu}$. Presumably, such a promiscuous usage of the glutamyl donors depends on the species of enzyme, but possibly TbtB and MibB may be able to use noncognate Glu-tRNA$^{Glu}$ from other species. More studies will reveal the intrinsic tolerance of this enzyme family to noncognate Glu-tRNA$^{Glu}$.

The FIT–GS system has facilitated insights into the structural motifs in the precursor peptide required for PTM in the GS biosynthesis. The knowledge obtained *in vitro* has been readily validated *in vivo*, allowing us to synthesize designer GS analogues in both *in vitro* and *in vivo*. Thus, GS would be an attractive scaffold to generate pseudo-natural products[35,36] for antibiotic or even other activities. Most importantly, we believe that the FIT system combined with various RiPP enzymes serves an excellent platform for the elaboration of PTM enzymes in other biosynthesis, facilitating the production of not only native secondary metabolites but also designer molecules.

## Methods

**General.** Oligonucleotides were purchased from Sigma-Aldrich. Synthetic genes for *godD* and GS-tandem were purchased from GENEWIZ (Supplementary Table 3). Ni-affinity chromatography for protein purification was performed on a Profinia (BioRad).

MALDI–TOF–MS analyses of FIT–GS products were performed on an ultrefrleXtreme (Bruker Daltonics). An ion trap MS system (amaZon SL; Bruker Daltonics) equipped with a high-performance liquid chromatography (HPLC) system (Agilent Technologies 1200 series, Agilent Technologies Japan, Ltd.) was used for LC–MS$^n$ analysis of enzymatically synthesized GS. A Cosmosil 5C$_{18}$ AR-II column (5 μm, 2.0 mm inside diameter × 150 mm length; Nacalai Tesque) was used for the analysis. Acetonitrile and 0.1% formic acid were used as eluents. The column temperature was kept at 40 °C and the flow rate was kept at 0.3 ml min$^{-1}$. The concentration of acetonitrile was kept at 5% for the first 2 min, then linearly increased to 95% over 25 min and at 95% for 5 min.

LC–MS analyses of serine/tyrosine-substituted GS analogues produced *in vivo* were performed on an ACQUITY UFLC (Waters) equipped with a Xevo G2-S Tof (Waters) operating in MS$^E$ mode (positive). An ACQUITY UFLC BEH C18 column (1.7 μm, 2.1 × 50 mm (Waters)) was used for the analyses. Acetonitrile and water containing 0.05% formic acid were used for eluents. The concentration of acetonitrile was set at 15%, then linearly increased to 85% over 10 min. The flow rate was 0.3 ml min$^{-1}$ and the column temperature was kept at 40 °C. Chromatograms were also monitored at 254 nm.

LC–MS analysis of GS-10SA11 and GS-tandem were performed on an ion trap system described above. In the case of GS-10SA11, a Cosmosil 2.5C$_{18}$ MS-II column (2.5 μm, 2.0 mm inside diameter × 100 mm length; Nacalai Tesque) was used for the analysis to check the production. Acetonitrile and water containing 0.1% formic acid were used as eluents. The column temperature was kept at 40 °C and the flow rate was kept at 0.4 ml min$^{-1}$. The concentration of acetonitrile was kept at 10% for the first 1 min, then linearly increased to 100% over 8 min and kept at 100% for 3 min. In the case of GS-tandem, a column and HPLC condition were identical to those for enzymatically synthesized GS. MS and MS$^2$ analyses of GS-10SA11 and GS-tandem were performed in ESI-positive mode. Chromatograms were also monitored at 254 nm.

HR–ESI–MS data of GS-10SA11 and GS-tandem were collected using a Bruker micrOTOF (Bruker Daltonics Inc., Massachusetts, USA) mass spectrometer in positive mode. One- and two-dimensional NMR spectra of GS-10SA11 were obtained on an ECA-600 (JEOL) at 600 MHz ($^1$H) or 150 MHz ($^{13}$C).

**Gene cloning.** Primers used in the gene cloning experiments were listed in Supplementary Table 4. *godD* sequence was optimized for protein expression in *E. coli* and amplified by PCR using pgodDopt-F-NdeI and pgodDopt-R-XhoI. *godE*, *godF* and *godH* were amplified by PCR using chromosomal DNA of *Streptomyces* sp. TP-A0584 as a template. *lazF* was amplified by PCR using pKU465-ltc18-6C (ref. 25) as a template. Primers, godE-N-Nde and godE-Cter-Hind were used for *godE*; godF-N-Nde and godF-C-Xho were used for *godF*; lazF-NdeI and lazF-R-XhoI were used for *lazF*; and godH-N-Nde and godH-C-Xho were used for *godH*. Amplified *godD* fragment was digested by NdeI and XhoI and cloned into the corresponding site of pET16b (Novagen) to give pET16-godD. *godF* and *godH* fragments were digested by NdeI and XhoI and cloned into the corresponding site of pET15b (Novagen) to give pET15-godF and pET15-godH, respectively. *godE* fragment was digested by NdeI and HindIII and cloned into the corresponding site of pET26b(+) (Novagen) to give pET26-godE. *lazF* was digested by NdeI and XhoI and cloned into the corresponding site of pET26b(+) to give pET26-lazF. After the sequencing, correct clones were selected for protein expression.

**Protein expression and purification.** For the expression of GodD, *E. coli* BL21(DE3) was transformed by pET16-godD and chaperon plasmid, pGro7 (Takara Bio Inc.). The transformant was inoculated into Luria-Bertani medium containing carbenicillin (50 μg ml$^{-1}$), chrolamphenicol (20 μg ml$^{-1}$) and L(+)-arabinose (0.5 mg ml$^{-1}$) and cultivated at 30 °C for 2 h. The temperature was reduced to 18 °C and the cells were grown at 18 °C for additional 20 h. The cells were collected by the centrifugation at 4,720*g* for 10 min and stored at −80 °C before further purification.

For the expression of GodE and LazF, *E. coli* BL21(DE3) was transformed by pET26-godE or pET26-lazF. The transformants were inoculated into ZYM-5052 medium[37] containing kanamycin (50 μg ml$^{-1}$) and cultivated at 18 °C for 20 h. The cells were collected by the centrifugation at 4,720*g* for 10 min and stored at −80 °C before further purification.

For the expression of GodF and GodH, *E. coli* BL21(DE3) was transformed by pET15-godF or pET15-godH. The transformants were inoculated into ZYM-5052 medium containing carbenicillin (50 μg ml$^{-1}$) and cultivated at 18 °C for 20 h. The cells were collected by the centrifugation at 4,720*g* for 10 min and stored at −80 °C before further purification.

For protein extraction, cells were suspended in the buffer containing 50 mM Tris-HCl pH8.0, 300 mM NaCl and 10 mM imidazole pH8.0. The cell suspension was sonicated by a UD-100 (TOMY). To separate the cellular debris from the soluble fraction, the lysate was centrifuged at 10,300*g* for 30 min at 4 °C. His-tagged

proteins were purified from the resulting supernatant by a Bio-Scale Mini Profinity IMAC Cartridge (Bio-Rad). The column was washed by the buffer containing 20 mM imidazole pH8.0, and then eluted by the same buffer containing 200 mM imidazole pH8.0. The eluted fraction was desalted by a Bio-Gel P-6 Desalting Cartridge (Bio-Rad) and the buffer was exchanged to 25 mM HEPES pH7.5 and 100 mM NaCl.

For the analyses of enzyme-bound co-factors, recombinant GodE, GodH and LazF were denatured by equal volume of methanol. After the centrifugation, the resulting supernatants were analysed by LC–MS. Flavin co-factors bound to recombinant GodE and LazF were detected by ESI-positive mode. Acetyl-CoA bound to recombinant GodH was detected by ESI-negative mode.

**Preparation of DNA templates for *in vitro* translation.** Double-stranded DNA templates that encode GodA mutants were prepared by primer extension and subsequent PCR. Primers used in this study are listed in Supplementary Table 5. Primer extension and PCR were performed using KOD -Plus- Ver.2 (TOYOBO Co., Ltd.). Concentrations of the buffer, dNTPs and MgSO$_4$ followed the supplier's instruction.

Appropriate forward and reverse primers were used for primer extension (details were listed in Supplementary Table 6). Each primer was mixed in PCR tube at the concentration of 1 μM. The primer extension was performed in 100 μl reaction mixture by denaturing at 95 °C for 1 min, followed by five cycles of annealing (50 °C for 1 min) and extending (68 °C for 1 min). A 5-μl aliquot of the reaction mixture was used as a template for subsequent PCR. In the first PCR, the template DNA was amplified using the designated forward and reverse primers (0.5 μM each). PCR was conducted in 100 μl reaction mixture by five cycles of denaturing (95 °C for 40 s) annealing (50 °C for 40 s) and extending (68 °C for 40 s). A 0.5-μl aliquot of the resulting PCR mixture was used as a template DNA for following PCR steps. The final PCR was conducted in 100 μl reaction mixture by 14 cycles of denaturing (95 °C for 40 s), annealing (50 °C for 40 s) and extending (68 °C for 40 s). Amplification of each PCR product was confirmed by agarose gel electrophoresis. The amplified DNA fragment was purified by FastGene Gel/PCR Extraction (NIPPON Genetics Co, Ltd.). DNA was eluted from the column by 10 μl of MilliQ water. Each eluent was directly used for *in vitro* translation reaction.

**Conditions for enzymatic reactions.** Translation factors, enzymes and ribosome were prepared and mixed as previously described to reconstitute an *in vitro* transcription–translation coupled system[22]. The reaction mixture contained final concentration of 50 mM HEPES · K (pH 7.6), 100 mM KOAc, 2 mM GTP, 2 mM ATP, 1 mM CTP, 1 mM UTP, 20 mM creatine phosphate, 12 mM Mg(OAc)$_2$, 2 mM spermidine, 2 mM DTT, 1.5 mg ml$^{-1}$ *E. coli* total tRNA (Roche), 1.2 μM ribosome, 0.6 μM MTF, 2.7 μM IF1, 0.4 μM IF2, 1.5 μM IF3, 30 μM EF-Tu, 30 μM EF-Ts, 0.26 μM EF-G, 0.25 μM RF2, 0.17 μM RF3, 0.5 μM RRF, 4 μg ml$^{-1}$ creatine kinase, 3 μg ml$^{-1}$ myokinase, 0.1 μM pyrophosphatase, 0.1 μM nucleotide-diphosphatase kinase, 0.1 μM T7 RNA polymerase, 0.73 μM AlaRS, 0.03 μM ArgRS, 0.38 μM AsnRS, 0.13 μM AspRS, 0.02 μM CysRS, 0.06 μM GlnRS, 0.23 μM GluRS, 0.09 μM GlyRS, 0.02 μM HisRS, 0.4 μM IleRS, 0.04 μM LeuRS, 0.11 μM LysRS, 0.03 μM MetRS, 0.68 μM PheRS, 0.16 μM ProRS, 0.04 μM SerRS, 0.09 μM ThrRS, 0.03 μM TrpRS, 0.02 μM TyrRS, 0.02 μM ValRS, 200 μM each proteinogenic amino acids and 100 μM 10-HCO-H$_4$folate.

Translation reaction was performed at 37 °C for 30 min in a 2.5-μl scale that contained 0.25 μl of DNA template. The resulting translation mixture (2.5 μl) was incubated with GodPTM enzymes (GodD, GodE, GodF and/or LazF; concentration of each enzyme was 1 μM) in the presence of 10 mM MgCl$_2$, 5 mM ATP and 1 mM DTT in a 15-μl scale. After the reaction was performed at 25 °C for 24 h, samples were desalted on a C-Tip C18 column (Nikkyo Technos) and analysed by MALDI–TOF–MS. Sinapinic acid/α-cyano-4-hydroxycinnamic acid was used as the matrix. All the samples were analysed by reflector mode, except for GodA*-tandem, which was analysed by linear mode. Calculated and observed *m/z* values are summarized in Supplementary Data.

For the total *in vitro* enzymatic synthesis of mature GS, GodA* was expressed in the *in vitro* translation system at 37 °C for 30 min in a 7.5 μl scale. The resulting translation mixture (7.5 μl) was incubated with GodPTM enzymes (GodD, GodE, GodF and/or LazF; concentration of each enzyme was 1 μM) in the presence of 10 mM MgCl$_2$, 5 mM ATP and 1 mM DTT in a 45 μl scale. A 15-μl aliquot was subjected to MALDI–TOF–MS analysis to confirm the proceeding of reactions. An amount of 1 μg of GluC (Roche Diagnostics K.K.) was then added to the remaining 30 μl aliquot and the resulting mixture was incubated at 37 °C for 1 h to digest the LP. After the digestion, GodH was added at the concentration of 6 μM and the resulting solution was incubated at 30 °C for 1 h. The reaction mixture was then lyophilized and dissolved in 100 μl of DMSO. The 25 μl of the DMSO solution was injected to LC–MS and the production of GS was analysed.

GS was selectively detected by MS$^3$ analysis in ESI-positive mode as follows. The precursor ion representing *m/z* 1,612 ± 2 was selected and fragmented. Then, resulting y16 fragment (*m/z* 1,319 ± 2) was further selected and fragmented.

**In vivo production of designer GS analogues.** Site-directed mutagenesis of *godA* and production of GS derivatives were performed as previously reported[3]. *Streptomyces* sp. TP-A0584 *ΔgodA* that harboured pGODR was used as a host

strain[31]. Primers for mutagenesis were listed in Supplementary Table 7. *n*-Butanol extracts of 7-day culture broth were analysed on an ACQUITY UFLC.

**Purification of GS-10SA11.** The *godA* disruptant of *Streptomyces* sp. TP-A0584 harbouring pTYM-10SA11 and pGODR was cultivated in V-22 (ref. 2) medium at 30 °C for 3 days. After the cultivation, a 3-ml portion of the seed culture was inoculated into 100 ml of GS medium (maltose 6.0%, pharma media 4.0%, yeast extract 1.0%, diaion HP20 1.0%, pH was adjusted to pH7.0) in a 500-ml K-1 flask and cultivated at 30 °C for 7 days.

Totally, 2.8 l of GS medium was prepared and extracted with an equal volume of *n*-butanol. The organic fraction was recovered and evaporated to give 11 g of crude sample. This fraction was resuspended in 500 ml of water and washed by *n*-hexane twice. Then, the aqueous fraction was recovered and extracted with an equal volume of ethyl acetate twice. Then the ethyl acetate fraction was recovered and evaporated to yield 1.2 g of crude sample. This sample was dissolved in a solution containing chloroform–methanol (10:1). The sample was subjected to LH-20 column chromatography and developed using the same solution. Fractions containing GS-10SA11 were collected and dried. As a result, 0.3 g of semi-pure GS-10SA11 was obtained. The sample was further purified by HPLC. A Protein-R (5 μm, 4.6 mm inside diameter × 250 mm length; Nacalai Tesque) was used for the purification. Acetonitrile and water containing 0.05% TFA were used for eluents. The concentration of acetonitrile was kept at 10% for the first 2 min, then linearly increased to 95% over 25 min and kept at 95% for 5 min. The flow rate was 1.5 ml min$^{-1}$ and the column temperature was kept at 40 °C. Chromatograms were monitored at 254 nm. Purification was repeated until 2.5 mg of pure GS-10SA11 was obtained.

**Production of GS-tandem.** Synthesized nucleotides encoding GS-tandem precursor peptide were cloned into NdeI-HindIII site of pTYM1gk vector. *Streptomyces* sp. TP-A0584 *ΔgodA* was transformed by the constructed plasmid. The resulting transformant was cultured in GS medium as described for GS-10SA11.

Totally, 2.9 l of GS medium was prepared and extracted by solvent containing chloroform–methanol (1:1). The organic fraction was recovered and evaporated. This fraction was resuspended in 90% methanol and washed by *n*-hexane. Then, the 90% methanol fraction was recovered and evaporated to give 2.2 g of crude sample. Then, the resultant residues were dissolved in 60% methanol and extracted by dichloromethane. The dichloromethane fraction was recovered and evaporated to yield 0.7 g of crude sample. This sample was dissolved in a solution containing chloroform–methanol (5:1). The sample was subjected to silica gel column chromatography and developed using the same solution. Fractions containing GS-tandem were collected and dried. This partially-purified sample was dissolved in a solution containing chloroform–methanol (10:1) and purified by LH-20 column chromatography developed by the same solution. The fractions containing GS-tandem were collected and evaporated to yield 22 mg of semi-pure sample. The sample was further purified by HPLC. A Protein-R (5 μm, 10 mm inside diameter × 250 mm length; Nacalai Tesque) was used for the purification. Acetonitrile and water containing 0.05% TFA were used for eluents. The concentration of acetonitrile was kept at 59% for the 15 min, then washed by 95% acetonitrile for 10 min. The flow rate was 3.0 ml min$^{-1}$ and the column temperature was kept at 40 °C. Chromatograms were monitored at 254 nm. Finally, 2.1 mg of GS-tandem was obtained.

**Data availability.** Data supporting the findings of this study are available within the article and its Supplementary Information files, and from the corresponding author upon reasonable request.

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

## Acknowledgements

We appreciate the assistance provided by Mizuho Nakaho and Yukari Kurokawa at Toyama Prefectural University, Hisashi Okamoto at The University of Tokyo for experiments, Yasuharu Kato at The University of Tokyo for technical supports in the *in vitro* experiments and Dr Joseph Rogers at The University of Tokyo for proof-reading. We also appreciate the LC–MS analysis support by Dr Mamoru Komatsu and Dr Kiyoko T. Miyamoto at Kitasato University, and Dr Naoya Oku at Toyama Prefectural University. This research was supported in part by a grant-in-aid from IFO, Institute for Fermentation, Osaka (to H.O., S.A. and T.O.), JSPS KAKENHI grant-in-aid for Young Scientists B (no. 26850043 to T.O. and no. 26850044 to S.A.), JSPS KAKENHI grant-in-aid for Challenging Exploratory Research (15K12739 to Y.G.), JST CREST of Molecular Technologies to H.S, JSPS KAKENHI (no. 25108707 to H.O.) and JSPS KAKENHI for Scientific Research on Innovative Areas (no. JP16H06444 to H.S, Y.G. and H.O.).

## Author contributions

T.O., S.A., Y.G., H.S. and H.O. designed the project and wrote the manuscript. T.O., K.Y., Y.G., S.A., Y.S., H.S. and H.O. designed the *in vitro* experiments, and K.Y. and M.S. performed the *in vitro* experiments. T.O., S.H., S.A., H.I. and H.O. contributed to the *in vivo* production of analogues and MS analyses. T.O., K.Y., Y.G., M.S., S.H., H.S. and H.O. analysed the data.

## Additional information

**Competing financial interests:** The authors declare no competing financial interests.

