## [Peer Review File · Nature Communications]

Reviewer #1 (Remarks to the Author):

In a large study Ozaki et al. report the one-pot synthesis of goadsporin (a 19 amino acid peptide containing 9 post-translational modifications) and dissection of its substrate recognition elements using >50 mutants. Although this peptide seems not particularly important, this is one of the most complicated PTM pathways reconstituted and has potential applications for creating novel active small molecules. I therefore recommend publication provided the presentation is greatly improved.

Title:

Understated and boring. How about

“Substrate recognition in goadsporin synthesis dissected by reconstitution of entire post-translational modification pathway”.

Introduction:

Does the peptide cross membranes?

Is 3D structure known? This is relevant to assess the importance of the structure of their variant GS-10SA11 determined in Figs S31-37.

Is the active part of the peptide known?

Has the pathway been reconstructed before in several pots?

Results:

For analysis of in vitro products only mass specs are presented in the figs and supplementary.

These say nothing about yields and specificities as it is easy to just pick out the peak you want and ignore the rest. I defer to someone better qualified in the field to assess their validity.

Chromatograms documenting yields should be presented (like in vivo Fig 6).

Discussion

For the variant they determined the structure of, is it active? Why was that variant and not the wild type selected for structure determination?

Line 394 states “GS would be an attractive scaffold” but the peptide is rather big to be crossing membranes and, based on the structure of their variant, circularization would presumably destroy its activity. The authors should discuss the exciting possibility that smaller scaffolds could be designed based on their data. This in turn requires a summary fig or table with the minimum recognition elements they determined for each of the 3 pathways boxed in Fig 1b.

The limitations of the method should be discussed.

Online methods

Line 642: instead of details of the reconstituted translation system, we are only given reference 21 which apparently focuses on flexizymes based on the ref 21 title. Presumably flexizymes are not involved but some (all?) aminoacyl-tRNA synthetases are? If so, the system is derived from Ueda's PURE system which should be stated and cited. Note that the upper box in Fig 1b (containing undefined translation proteins and one tRNA) is unhelpful on this question.

Lines 646, 656, 659 stated PTM enzymes were used at 1 uM, 1 ug (concentration?) and 6 uM; line 249 stated “stoichiometric or excess amounts of PTM enzymes”; line 337 stated “comparatively high concentrations of PTM enzymes”. Were those high amounts necessary; i.e. what were the dose-response curves used for optimization? This is important for engineering; e.g. does using too much enzyme cause side reactions?

Reviewer #2 (Remarks to the Author):

Goadsporin belongs to the RiPPs family of natural products and is generated from a ribosomally produced precursor peptide. The authors employ an in vitro translation system to produce

precursor peptides and add purified goadsporin biosynthetic enzymes to convert the precursor peptides to the natural product and several derivatives. Ozaki et al. then utilize this system for a comprehensive analysis of the substrate tolerance and enzymatic specificities of the enzymes from the goadsporin pathway. The insights are successfully implemented in an in vivo system to produce larger quantities of goadsporing derivatives than the in vitro system would be able to offer.

This very comprehensive work is the logical extension of previous experiments, in which one of the corresponding authors used an in vitro translation system to analyze an enzyme from the Patellamide pathway. The manuscript is reasonably well written (see below) and not only provides new and detailed insights into the biosynthesis of goadsporin but also offers a methodology to study other RiPPs pathways.

This reviewer is of the opinion that the manuscript should be accepted for publication once the authors have addressed the following points:

1. The lactazole biosynthetic gene cluster is shown in Figure 1A but mentioned for the first time much later in the manuscript. This is confusing and should be remedied.
2. The authors report that the precursor peptide GodA* contains an internal disulfide bond (Line 115), but do not state how it was removed for further experiments. At the moment it reads as if the peptide was used as is, and that the disulfide is somehow removed by the PTMs.
3. The authors claim that the use of GodA*-tandem is rationally designed based on data from their in vitro system. This reviewer fails to see that connection. The result is important and interesting, reminiscent of other RiPPs pathways in which one precursor peptide contains multiple core peptides. But how was it rationally designed?

Minor issues:

The figure legends of S5 – S16 have inconsistent font sizes.

There are many instances in which the writing could be improved. Here are a few:

Line 49: should read GS-resistance

Line 72: should read involved in RiPPs biosynthesis

Line 83: should read with the posttranslational

Line 85: should read referred to as the FIT-PatD system

Line 86: should read enabled the expression

Line 87 should read PatD-catalyzed

Line 89: should read FIT-PatD system revealed the

Line 90/91: should read sequences of the substrate

Line 92: should read particular work exploited only the in vitro activity

Line 95: should read referred to as the FIT-GS system

Line 96: should read produce native GS

Line 98/99: should read by means of the FIT-GS system

Line 100/101: should read , we were able to produce designer GS derivatives in vivo as well as in vitro, whose...

Line 114/115: should read corresponds to GodA*

Line 140: should read GodE was also included

Line 239: should read two Ala point-mutants

Line 260: pick either entire or whole

Line 279: should read leading to the formation of an additional azole.

Line 291: should read encoding the transcriptional activator of the the GS biosynthetic

Line 308: should read we tested the in vivo expression

Line 313: should read we then expressed the designer GodA-10SA11 in vivo.

Line 326: should read from a synthetic DNA

Reviewer #3 (Remarks to the Author):

The ms. entitled "Translation-coupled enzymatic synthesis of goadsporin and its analogs in vitro and in vivo" by Ozaki et al describes interesting experiments on the in vitro biosynthesis of a ribosomally-produced peptides.

While the concept is not new, the authors show a highly interesting collection of results as they show an in-depth analysis of how the (otherwise well established) FIT-technology can be applied to the particular problem set. Ribosomally-produced peptides are frequently of high bioactivity and hard to synthesize using conventional organic synthesis - the engineered biosynthesis thus if of high relevance. While it is often difficult to isolate the peptides from heterologous fermentation experiments (e.g. in *E. coli* or *S. cerevisiae*), the full in vitro approach appears to be more efficient when it comes to quickly access the targeted cpd. Here, the authors show, how this approach can be used for a targeted library creation.

Overall, I consider the potential impact of this ms to be very significant and the claims appear justified.

To Reviewer #1

We are grateful to Reviewer #1 for the favorable and valid comments regarding our manuscript.

Title:

Understated and boring. How about "Substrate recognition in goadsporin synthesis dissected by reconstitution of the entire post-translational modification pathway".

On the basis of your suggested title, albeit not the exact title, we change it to " *Dissection of goadsporin biosynthesis by in vitro reconstitution leading to designer analogs expressed in vivo* "

Introduction:

Does the peptide cross membranes?

Unfortunately, we have no direct evidences that GS (or GS analogs) crosses membranes. However we identified that GS can bind to Signal Recognition Particle (SRP) which localized in the cell cytoplasm (unpublished data), and self-resistance gene of goadsporin is *ffh* homolog which is a component of SRP as mentioned in line 47. The exogenous addition of GS to the other *Streptomyces* are also effective. Therefore, we believe that GS is likely able to cross the membranes.

Is 3D structure known? This is relevant to assess the importance of the structure of their variant GS-10SA11 determined in Figs S31-37.

Is the active part of the peptide known?

The 3D structure of GS is currently unknown and the critical activity determinant of the GS bioactivity has not been yet determined. One thing, however, currently known is that two dehydroalanine residues seems essential for its bioactivity, because goadsporin B, a GS analog missing the dehydroalanines, does not have the same activity as GS (T. Ozaki *et al.* Chembiochem, 2016). Even though the current study did not include bioactivity tests for GS analogs, more systematic designs of GS analogs with bioactivity tests should reveal such determinants in the future.

Has the pathway been reconstructed before in several pots?

We did not clearly understand the question, but we assume the question as asking us whether any step of the GS PTMs had been reconstituted previously. The answer is 'no'.

Results:

For analysis of in vitro products only mass specs are presented in the figs and supplementary. These say nothing about yields and specificities as it is easy to just pick out the peak you want and ignore the rest. I defer to someone better qualified in the field to assess their validity. Chromatograms documenting yields should be presented (like in vivo Fig 6).

The virtue of this work is to reconstitute the individual steps of the GodPTMs, which enables us to rapidly dissect the biosynthesis pathway and qualitatively analyze the product formed in a small scale (2.5 μ L translation scale), *i.e.* inexpensively. In fact, even though the MALDI-TOF data are qualitative, they are quite informative for synthesis of designer GSs. Moreover, the LC-MS

chromatograms shown in Fig 2f of the original manuscript enabled us to estimate a concentration of the *in vitro* synthesized GS to be an approximately 7 nM by comparison with the chromatogram of the authentic GS. To clarify the yield of GS in the *in vitro* system, we have added the description in line 170–174 as follows;

The observed peak areas of GS expressed in the FIT-GS system allowed for estimating a concentration to be an approximately 7 nM in comparison with known quantities of the authentic GS samples. These results explicitly showed the *in vitro* synthesis of the desired GS in the FIT-GS system.

We also added the following comment in the legend of Figure 2f.
(62.5 pg of authentic GS was injected for this chromatogram)

Discussion

For the variant they determined the structure of, is it active? Why was that variant and not the wild type selected for structure determination?

We have not examined the activity of the synthesized GS analogs in the present study, but we are certainly interested in testing the bioactivity with more variety of analogs in the future.

The second question was unfortunately unclear to us. However it should be noted that the structure determination of GS had been already described in our previous paper (Y. Igarashi *et al.* J. Antibiot., 2001); therefore, we did not describe the structural determination of wild type GS in this article.

Line 394 states “GS would be an attractive scaffold” but the peptide is rather big to be crossing membranes and, based on the structure of their variant, circularization would presumably destroy its activity. The authors should discuss the exciting possibility that smaller scaffolds could be designed based on their data.

As mentioned earlier, we do not have direct evidence if GS is membrane permeable, but it could be. However, we agree with this reviewer that it is very interesting to discuss the potentials of smaller scaffolds. We added in line 356–363 as follows:

According to this principle, we have demonstrated the expression of various lengths ofazole-containing peptides consisting of the X₁-S/T/C-X₂ motif ranging from a single to 12 motifs in the FIT-GS system (Fig. 3a, entries 2–4, 10–13, 20–30, and 35–37, and Fig3d, entries 54 and 55). To the best of our knowledge, such short (3 residues)azole-containing RiPPs have not yet been discovered in nature. However, the successful *in vitro* expression of such short RiPPs will allow us to investigate not only a possibility of their *in vivo* expression but also their potentials for bioactivities.

This in turn requires a summary fig or table with the minimum recognition elements they determined for each of the 3 pathways boxed in Fig 1b.

According to the reviewer’s suggestion, we added Figure 7 for a summary. A legend for this figure was also added.

Figure 7. Sequence requirements for GodPTMs dissected in this study.

Through examining various analogs both *in vitro* and *in vivo*, sequence requirements for GodPTMs were proposed. The C-terminal 19 residues highlighted in red in the LP region were sufficient for the formation of azoles and Dhas. In the azole formation catalyzed by GodD/E, X1-S/T/C-X2 motif could be the recognition determinant. The length of the central region in the core peptide can be varied from one to three. Dha formation required preceding azole formation. Ser4 would be modified prior to Ser14.

The limitations of the method should be discussed.

According to the comment, the sentence in line 338 was changed as follows;

Due to the economical reason, this *in vitro* system is suitable for qualitative analysis in small scales, not for quantitative analyses. However, it enables us to access the expression of a wide array of mutants and designer analogs simply by *in vitro* preparation of the corresponding DNA templates. This advantage significantly reduces the labors of chemical synthesis or bacterial expression/isolation of precursor peptides previously described.^{30,32}

Online methods

Line 642: instead of details of the reconstituted translation system, we are only given reference 21 which apparently focuses on flexizymes based on the ref 21 title. Presumably flexizymes are not involved but some (all?) aminoacyl-tRNA synthetases are? If so, the system is derived from Ueda's PURE system which should be stated and cited. Note that the upper box in Fig 1b (containing undefined translation proteins and one tRNA) is unhelpful on this question.

Although the title of the Nature Protocols paper (ref 22) focuses on the flexizyme technology, it is indeed the first paper where the term of "FIT system" is introduced and also the paper providing the most detailed description of the components and conditions of the FIT system; thus we believe this citation is appropriate. The FIT system is based on a reconstituted *in vitro* translation system developed by Prof. Ueda's group (so-called PURE system), but the concentrations of the translation factors and other ingredients have been optimized for the expression of peptides, not proteins, described in this article. In this report the FIT system contains all aminoacyl-tRNA synthetases as well as all *E. coli* tRNAs, so that to clarify these issues, we newly added the Ueda's PURE system paper into the reference list and revised the figure 1b, the main text, and the supporting information as follows;

(Main text: line 82) We previously devised an *in vitro* reconstituted biosynthetic system²¹ for the synthesis of various azoline-containing peptides by combination of a custom-made cell-free translation (Flexible In-vitro Translation; FIT²²) system composed of reconstituted translation components²³ with the posttranslational cyclodehydratase PatD.⁹

(Online methods: Page 35, line 657) The reaction mixture contained final concentration of 50 mM HEPES-K (pH 7.6), 100 mM KOAc, 2 mM GTP, 2 mM ATP, 1 mM CTP, 1 mM UTP, 20 mM creatine phosphate, 12 mM Mg(OAc)₂, 2 mM spermidine, 2 mM DTT, 1.5 mg/mL *E. coli* total tRNA (Roche), 1.2 μM ribosome, 0.6 μM MTF, 2.7 μM IF1, 0.4 μM IF2, 1.5 μM IF3, 30 μM EF-Tu, 30 μM EF-Ts, 0.26 μM EF-G, 0.25 μM RF2, 0.17 μM RF3, 0.5 μM RRF, 4 μg/mL creatine kinase, 3 μg/mL myokinase, 0.1 μM pyrophosphatase, 0.1 μM nucleotide-diphosphatase kinase, 0.1 μM T7 RNA polymerase, 0.73 μM AlaRS, 0.03 μM ArgRS, 0.38 μM AsnRS, 0.13 μM AspRS,

0.02 μM CysRS, 0.06 μM GlnRS, 0.23 μM GluRS, 0.09 μM GlyRS, 0.02 μM HisRS, 0.4 μM IleRS, 0.04 μM LeuRS, 0.11 μM LysRS, 0.03 μM MetRS, 0.68 μM PheRS, 0.16 μM ProRS, 0.04 μM SerRS, 0.09 μM ThrRS, 0.03 μM TrpRS, 0.02 μM TyrRS, 0.02 μM ValRS, 200 μM each proteinogenic amino acids, and 100 μM 10-HCO-H₄folate.

Lines 646, 656, 659 stated PTM enzymes were used at 1 μM , 1 μg (concentration?) and 6 μM ; line 249 stated “stoichiometric or excess amounts of PTM enzymes”; line 337 stated “comparatively high concentrations of PTM enzymes”. Were those high amounts necessary; i.e. what were the dose-response curves used for optimization? This is important for engineering; e.g. does using too much enzyme cause side reactions?

Because MALDI-TOF-MS analyses do not provide quantitative data, we did not check the dose-response curves. However, in our preliminary experiments, several concentrations of GodD/E were tested to optimize the reaction condition. Concentrations over 0.2 μM gave a single reaction product containing six azoles (Review-only Material attached below, **d** and **e**), whereas lower enzyme concentration resulted in incomplete GodA* consumption and/or production of partially cyclodehydrated products (Review-only Material, **b** and **c**). To examine the conversion of less reactive GodA* analogs, 1 μM was finally chosen for further experiments. Albeit high concentrations of the PTM enzymes, unexpected side reactions were not observed in the present study. As mentioned in line 350, we believe that our *in vitro* results would be relevant because they were consistent with *in vivo* results.

Reviewer #2 (Remarks to the Author):

Goadsporin belongs to the RiPPs family of natural products and is generated from a ribosomally produced precursor peptide. The authors employ an in vitro translation system to produce precursor peptides and add purified goadsporin biosynthetic enzymes to convert the precursor peptides to the natural product and several derivatives. Ozaki et al. then utilize this system for a comprehensive analysis of the substrate tolerance and enzymatic specificities of the enzymes from the goadsporin pathway. The insights are successfully implemented in an in vivo system to produce larger quantities of goadsporing derivatives than the in vitro system would be able to offer. This very comprehensive work is the logical extension of previous experiments, in which one of the corresponding authors used an in vitro translation system to analyze an enzyme from the Patellamide pathway. The manuscript is reasonably well written (see below) and not only provides new and detailed insights into the biosynthesis of goadsporin but also offers a methodology to study other RiPPs pathways.

This reviewer is of the opinion that the manuscript should be accepted for publication once the authors have addressed the following points:

We are grateful to Reviewer #2 for the favorable and valid comments regarding our manuscript.

1. The lactazole biosynthetic gene cluster is shown in Figure 1A but mentioned for the first time much later in the manuscript. This is confusing and should be remedied.

As following the reviewer's suggestion, to enhance readability, we deleted the organization map of the lactazole biosynthetic gene cluster in Fig.1 and moved it to Supplementary Fig. 3. According to this change, the figure numbers were changed appropriately.

2. The authors report that the precursor peptide GodA contains an internal disulfide bond (Line 115), but do not state how it was removed for further experiments. At the moment it reads as if the peptide was used as is, and that the disulfide is somehow removed by the PTMs.*

In the following PTM reaction mixture, 1 mM of DTT was included as described in the method section. Therefore, the internal disulfide on GodA* should be chemically reduced prior to the PTMs. To clarify this issue, we mentioned the addition of DTT also in the main text in the revised manuscript (line 116) as follows:

GodA* was used in the presence of 1 mM dithiothreitol (DTT) to reduce the disulfide bond in the following studies with the reconstituted GodPTM enzymes.

3. The authors claim that the use of GodA-tandem is rationally designed based on data from their in vitro system. This reviewer fails to see that connection. The result is important and interesting, reminiscent of other RiPPs pathways in which one precursor peptide contains multiple core peptides. But how was it rationally designed?*

This reviewer seems misunderstood that the GodA*-tandem consists of dual core peptides, i.e. two GS molecules from GodA*-tandem are produced. However, this analog has a single core peptide consisting of 38-mer continuous residues. Nonetheless, we thank this comment and agree

that the design of this analog was not clearly described in the original manuscript. Thus, we have added detailed descriptions in line 286 as follows:

Since GodA*-10SA11 was modified and relevance of an X₁-S/T/C-X₂ motif was validated, we conceived that a longer GS analog could be synthesized in a similar manner. To demonstrate this idea, we designed GodA*-tandem that has a 38-mer core peptide composed of tandemly repeated the original core sequence of GodA and expressed it in the FIT-GodD/E system (Fig. 3d, entry 55)."

Minor issues:

The figure legends of S5 – S16 have inconsistent font sizes.

We changed font size larger in S5 to S16

There are many instances in which the writing could be improved. Here are a few:

Line 49: should read GS-resistance

Line 72: should read involved in RiPPs biosynthesis

Line 83: should read with the posttranslational

Line 85: should read referred to as the FIT-PatD system

Line 86: should read enabled the expression

Line 87 should read PatD-catalyzed

Line 89: should read FIT-PatD system revealed the

Line 90/91: should read sequences of the substrate

Line 92: should read particular work exploited only the in vitro activity

Line 95: should read referred to as the FIT-GS system

Line 96: should read produce native GS

Line 98/99: should read by means of the FIT-GS system

Line 100/101: should read , we were able to produce designer GS derivatives in vivo as well as in vitro, whose...

Line 114/115: should read corresponds to GodA*

Line 140: should read GodE was also included

Line 239: should read two Ala point-mutants

Line 260: pick either entire or whole

Line 279: should read leading to the formation of an additional azole.

Line 291: should read encoding the transcriptional activator of the the GS biosynthetic

Line 308: should read we tested the in vivo expression

Line 313: should read we then expressed the designer GodA-10SA11 in vivo.

Line 326: should read from a synthetic DNA

We thank you for the suggestions and implemented them into the revised manuscript.

Reviewer #3 (Remarks to the Author):

The ms. entitled "Translation-coupled enzymatic synthesis of goadsporin and its analogs in vitro and in vivo" by Ozaki et al describes interesting experiments on the in vitro biosynthesis of a ribosomally-produced peptides.

While the concept is not new, the authors show a highly interesting collection of results as they show an in-depth analysis of how the (otherwise well established) FIT-technology can be applied to

the particular problem set. Ribosomally-produced peptides are frequently of high bioactivity and hard to synthesize using conventional organic synthesis - the engineered biosynthesis thus if of high relevance. While it is often difficult to isolate the peptides from heterologous fermentation experiments (e.g. in *E. coli* or *S. cerevisiae*), the full in vitro approach appears to be more efficient when it comes to quickly access the targeted cpd. Here, the authors show, how this approach can be used for a targeted library creation.

Overall, I consider the potential impact of this ms to be very significant and the claims appear justified.

We thank this reviewer for the supportive comments.

Reviewer #1 (Remarks to the Author):

The new title is much better, other comments of the 3 reviewers are addressed well and the manuscript should now be accepted for publishing.

Minor comments for addressing in proofs:

Introduction:

To answer original reviewer request, state that GS is likely to cross the membranes and its 3D structure is unknown.

New summary Fig 7:

"C-terminal" typo. Also, as a stand alone summary fig it should be clarified. The sequences referred to by "Leader peptide" and "Core peptide" should be indicated with brackets below these words.

Also "C-terminal 19 residues are" should be C-terminal 19 residues of the leader peptide are (to distinguish from the real C-terminal 19 residues that exactly correspond to the core peptide).

Supp: The review-only material is nice and may be added to the supp if the authors so desire.

Reviewer #2 (Remarks to the Author):

The paper has been improved and can now be accepted.

Responses to the reviewer #1

We are grateful to Reviewer #1 for the valid comments regarding our manuscript.

Introduction:

To answer original reviewer request, state that GS is likely to cross the membranes and its 3D structure is unknown.

As the reviewer's suggestion, we revised manuscript (line 47) as follows: “Although the 3D structure is unknown, GS very likely crosses the cell membrane and targets the Ffh protein ...”

New summary Fig 7:

"C-teminal" typo. Also, as a stand alone summary fig it should be clarified. The sequences referred to by "Leader peptide" and "Core peptide" should be indicated with brackets below these words. Also "C-teminal 19 residues are" should be C-terminal 19 residues of the leader peptide are (to distinguish from the real C-terminal 19 residues that exactly correspond to the core peptide).

This Figure was renumbered as Figure 10 due to other changes requested in editorial processes. We indicated amino acid sequences of "Leader peptide" and "Core peptide" with brackets in Fig. 10 following the reviewer's suggestion.

Supp: The review-only material is nice and may be added to the supp if the authors so desire.

Thank you for your comment, but we think that the review-only material is not necessary in SI.

Responses to the reviewer #2

We thank this reviewer for the supportive comments.

Responses to the format changes requested by the editor

Main text

As requested, we shortened the title within 15 words as “Dissection of goadsporin biosynthesis by in vitro reconstitution leading to designer analogs expressed in vivo”.

According to the suggestion, abstract was revised to introduce the background and context of the work before describing the key findings of the present study.

Subheadings of the method sections that exceed the word-length limit were shortened as follows;

Page 29, line 504 “Preparation of DNA templates for in vitro translation”

Page 33, line 568 “ In vivo production of designer GS analogs”

Page 7, line 106, “Fig. S1” was changed to “Supplementary Fig. 1”. Same changes were applied to all references to supplementary materials throughout the main text.

According to the suggestion, the phrases, (data not shown) were removed from page 7 line 110 and page 8 line 123.

The style of references were changed to Nature style. DOI was removed from the references. Methods-only reference (ref. 37) was listed with other references.

As suggested, online methods were moved to Methods section in the main text.

Following data availability statement was added after Methods section;

“Data availability. Data supporting the findings of this study are available within the article and its supplementary information files, and from the corresponding author upon reasonable request.”

Following financial competing interests statement was added after References;

“Competing financial interests: The authors declare no competing financial interests.”

The titles of Figures 1 and 5 were shortened to fit into the style.

All the chemical structures were also uploaded as separate Chemdraw files as requested.

As suggested, the former Figure 3 was split into separate Figures (Figure 3, Figure 5, Figure 6 and Figure 7). To maintain the current style emphasizing each residue with colored graphics, we would like to provide these items as Figures rather than Tables as you suggested. We believe it will help the readers to visually understand the results.

According to this change, figure legends were changed appropriately.

In addition to the above editorial changes, the names and the sequences of the cloning primers for *godF* and *godH* in Supplementary Table 4 were corrected since we mentioned different primers in error. Corresponding part in the Methods section was changed appropriately. Sizes of the molecular weight marker in Figure S2 were also corrected. These changes do not affect the results of the manuscript.

Supplementary information

The title page and author list were removed as requested.

Because the former Supplementary Table 5 is too large to be embedded in the supplementary information file, we decided to provide this as a separate Supplementary Data.

Other Supplementary Tables were incorporated in the SI file and listed in numerical order. Since Online Methods was changed to Methods section, the supplementary tables were re-numbered properly. During this process, Supplementary Table 8 and Supplementary Table 9 in the previous version were combined and as Supplementary Table 3 in the revised manuscript.

According to these changes, references to Supplementary Tables in the main text were changed appropriately.

Editor's Summary

We thank you for the suggestion for Editor's Summary, but made a little modification to your version. Please consider our revised summary.

Featured Image

Featured image was uploaded.

A completed author checklist, completed and signed copies of image License to Publish (LTP) for any Featured Image suggestions, and completed and signed copies of open access publication forms for the Article were provided with manuscripts.